# Backward induction-based deep image search

**Donghwan Lee** , **Wooju Kim** *

Department of Industrial Engineering, Yonsei University, Seoul, Republic of Korea

* wkim@yonsei.ac.kr

## Abstract

Conditional image retrieval (CIR), which involves retrieving images by a query image along with user-specified conditions, is essential in computer vision research for efficient image search and automated image analysis. The existing approaches, such as composed image retrieval (CoIR) methods, have been actively studied. However, these methods face challenges as they require either a triplet dataset or richly annotated image-text pairs, which are expensive to obtain. In this work, we demonstrate that CIR at the image-level concept can be achieved using an inverse mapping approach that explores the model's inductive knowledge. Our proposed CIR method, called Backward Search, updates the query embedding to conform to the condition. Specifically, the embedding of the query image is updated by predicting the probability of the label and minimizing the difference from the condition label. This enables CIR with image-level concepts while preserving the context of the query. In this paper, we introduce the Backward Search method that enables single and multi-conditional image retrieval. Moreover, we efficiently reduce the computation time by distilling the knowledge. We conduct experiments using the WikiArt, aPY, and CUB benchmark datasets. The proposed method achieves an average mAP@10 of 0.541 on the datasets, demonstrating a marked improvement compared to the CoIR methods in our comparative experiments. Furthermore, by employing knowledge distillation with the Backward Search model as the teacher, the student model achieves a significant reduction in computation time, up to 160 times faster with only a slight decrease in performance. The implementation of our method is available at the following URL: https://github.com/dhlee-work/BackwardSearch.

**Data Availability Statement:** All datasets are available from the follow URLs: Wikiart dataset : https://huggingface.co/datasets/huggan/wikiart CUB dataset : https://www.vision.caltech.edu/datasets/cub_200_2011/ aPY dataset : https://vision.cs.uiuc.edu/attributes/.

## Introduction

Image retrieval (IR) is a critical research area in computer vision [1, 2]. One of its subareas, content-based IR (CBIR), retrieves images similar to a query image from the database. By performing visual analysis directly, the CBIR method enhances the quality of IR without requiring the complete annotation of images in the database [3]. In traditional CBIR, image features are manually extracted and the similarity between images is determined by calculating their distances [4–6]. These handcraft methods have the benefit of feature vector interpretability [7].

The convolutional neural network (CNN) learns features of two-dimensional images automatically and effectively [8, 9]. Compared to the handcrafted feature extraction method, the

**Funding:** The author(s) received no specific funding for this work.

**Competing interests:** The authors have declared that no competing interests exist.

use of CNN for CBIR has significantly improved the performance of IR [10, 11]. As a lower layer of the CNN extracting low-level features, the deeper layers enable extract semantic features from images [11, 12]. The features extracted from the model are used to perform high-quality IR. When the latent features of an image are smoothly transitioned on the latent space, the corresponding image also changes [13–15]. This trend is also observed when the model, which is trained with the ImageNet dataset, is adopted in another domain dataset (e.g., a digitized painting dataset) [16, 17]. The CBIR method employs these characteristics [18].

In CBIR applications, reflecting the user search intention to a large extent is generally challenging. Furthermore, retrieving the same image with different query images occurs frequently. This problem arises from the "hubness problem" [19], where the center point on the latent space tends to be extracted from many different embedding vectors. Conditional IR (CIR) is an alternative method to increase the diversity of the retrieval results by specifying the search scope and determining the database to be searched. However, conventional CIR methods require a fully annotated label [16].

Composed image retrieval (CoIR) [20–22] is a method for CIR. The existing CoIR models combine embeddings of query images and conditioning text utilizing a fusion module. Although the fusion of two different modal embeddings has shown great success, these CoIR methods require lots of triplets <reference image, conditioning text, target image> dataset which is expensive to collect and verify for training the model. Recently, A zero-shot CoIR (ZS-CoIR) approach has been studied for generalizability. ZS-CoIR methods are trained using only image-text pairs [23, 24] or automatically annotated triplets [25, 26]. Although being successfully evaluated on the FashionIQ [27] and CIRR [28] datasets, ZS-CoIR requires richly detailed image-level text descriptions and a large amount of such data, to achieve high model performance.

In this study, we demonstrate that CIR at the image label can be achieved through an inverse mapping approach by leveraging the model's inductive knowledge while maintaining the query's context. Specifically, the proposed approach consists of a query image encoder and the corresponding label mapping function ϕ to learn the relationship between a query and its associated label. In the course of a conditional search, as the embedding vector of a query image and the trained ϕ is prepared, backpropagation is used to update the query's embedding ensuring that the ϕ generates the corresponding condition label. During the process, of Backward Search, the embedding of the query is iteratively updated to an embedding vector which ϕ generates a conditioned label. As illustrated in Fig 1 for visual understanding, the latent vector (a) of the "Impressionism" query image is iteratively updated to a point (c) following the explicit condition "Rococo" via backpropagation. From (a) to (c), the updated vector passes through (b), which is the transitional route.

Based on the method described above, we introduce a Backward Search method at the image-level labels that enables both single and multi-conditional image retrieval while preserving the contextual information of the query image, even with models not trained on triplet relationships or rich textual annotations. However, the proposed method requires several iterations in the Backward Search process. To reduce time consumption, we employ a knowledge distillation method that simply minimizes the mean squared error (MSE) of the output logits between the student model and the proposed Backward Search process. Using the proposed method, images can be effectively searched based on the user's intent from large image databases with complex contextual information, even when only a portion of the images are annotated with labels.

The WikiArt [29], aPY [30], and CUB [31] datasets were used in this study. An ablation study was performed and t-SNE was employed for visualizing the characteristic of the proposed Backward Search algorithm on the embedding space. In the comparative study, to the

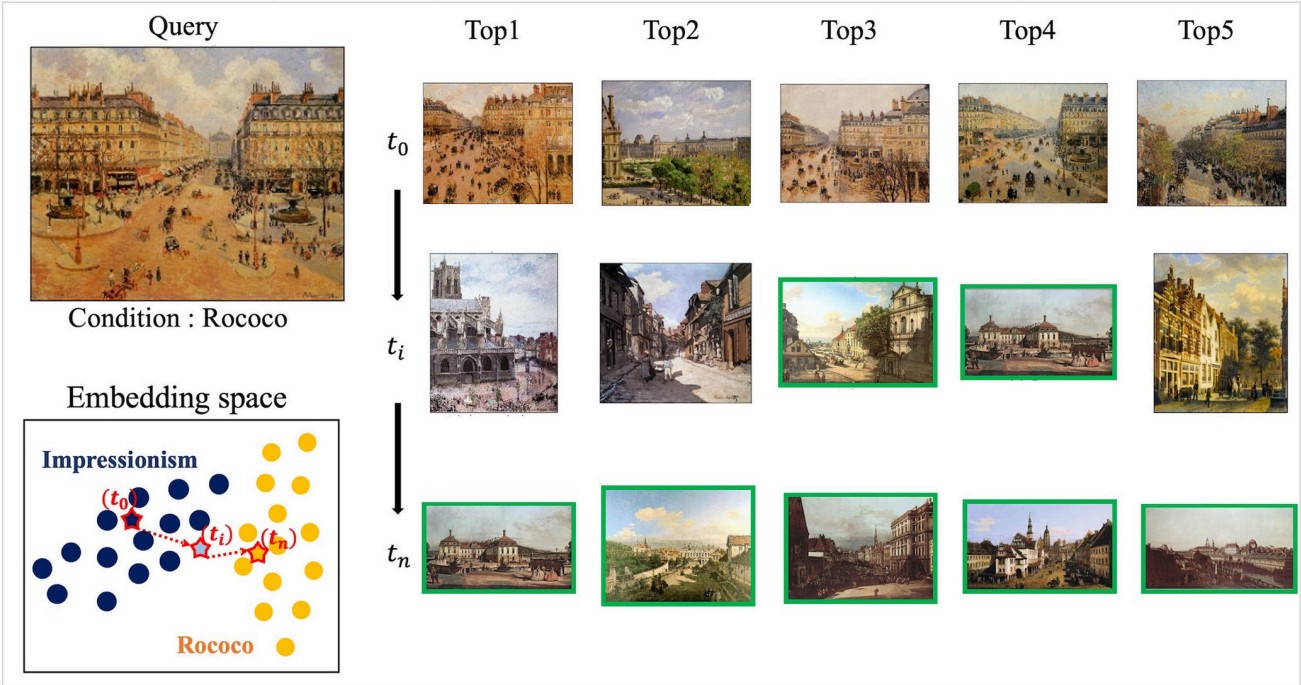

**Fig 1. Overview of the proposed method.** Example query from the WikiArt dataset, where $t_o$ is initial IR results, and $t_n$ is Rococo' conditioned IR results. The proposed method updates embedding vectors $t_o$ to $t_n$ through $t_i$ by exploring the embedding space. A green box indicates that the retrieved image has the same class as the condition.

best of our knowledge, there has been no study on deep learning-based conditional image search at the image-level label. Thus, although there are limitations in interpreting the results, we compared our model with those in a similar task. The CoIR models were used for comparative experiments, and the results show an average mAP@10 of 0.541 on the datasets, demonstrating competitive performance compared to other models. Moreover, by employing knowledge distillation with the Backward Search model, the student model achieves a significant reduction in computation time, up to 160 times faster, with only a minimal drop in performance.

The contributions of this study are as follows: (1) We propose a conditional image retrieval method, Backward Search, using a model only trained only on a pair of images and image-level labels by leveraging inductive knowledge. (2) The proposed method enables single and multi-conditional image retrieval that satisfies specified conditions while preserving the contextual information of the query image. (3) We efficiently reduce the computation time by distilling the knowledge from the Backward Search process into a small forward neural network. (4) Large image databases with complex context can be effectively searched based on the user's intent, even when only a portion of the images are annotated with labels. (5) We release corrected attributions for digitized artwork images in the WikiArt dataset.

## Related work

Deep learning-based CBIR methods focus on learning image feature representations and constructing a well-organized embedding space for a dataset. Pretrained models learned from extensive image datasets are commonly employed as backbone models in CBIR tasks to

achieve this aim. In early studies using deep neural networks, Krizhevsky et al. [32] attempted to learn embedding vectors as a feature for IR using an autoencoder, and Xia et al. [33] used class labels to learn the representation of a CNN model, achieving promising IR performance. In a recent study, Zhao et al. [34] established fine-tuned classification models trained with painting images, demonstrating that the fine-tuned model can conduct IR. Kiran et al. [35] proposed a deep learning-based model that captures channel-wise, low-level features of an image using a sparse autoencoder and a VGG-16 model. Moreover, Hamilton et al. [16] found that extracted features with ImageNet pre-trained models can search similar images without additional dimension reduction.

Hamilton et al. [16] introduced a CIR algorithm by pruning irrelevant label nodes within a predefined k-d tree. However, the algorithm requires a fully annotated label for the training and test datasets. The CoIR method, which is part of CIR, has been actively researched [20–26]. For an ordinary CoIR method, Baldrati et al. [22] proposed CLIP4CIR, which includes a combiner module to fuse embeddings of a reference image, and text conditioning using fine-tuned CLIP [36] encoders to predict target image embeddings. To improve the generalization of CoIR, a zero-shot method has been actively studied, Saito et al. [23] introduced Pic2Word, which uses only image-text pairs based on pre-trained CLIP. Ventura et al. [25] CoVR trained automatically generated triplets of video datasets. Baldrati et al. [24] proposed SEARL, which learns from only unlabeled images and generated captions from GPT guidance. In particular, the SEARLE is a zero-shot model designed to maintain image-level concepts while transforming contextual information. The SEARLE model uses GPT to generate accurate text descriptions of the image's concept from unlabeled images for training. While it seems similar to our proposed method in its use of inversion and Knowledge Distillation (KD) techniques, the SEARLE model employs inversion to obtain concept embeddings of images and simplifies this process using KD. These ZS-CoIR methods have shown promising performance even with unseen data such as FashionIQ [27] and CIRR [28].

The inversion approach used in the proposed Backward Search method involves using the outcomes of actual observations to deduce the parameter values characterizing the system and estimate data not readily observable directly [37]. In the context of images, deep neural networks have successfully addressed inverse problems, including image recovery, restoration, deconvolution, super-resolution, anomaly detection, and others [38, 39]. Considering that most methods commonly use a regularization term to alleviate ill-posed behavior [40], the proposed method uses regularization to preserve the features of the query image.

Knowledge distillation (KD) has been applied to reduce the computational cost of Backward Search. Knowledge distillation aims to transfer knowledge from a cumbersome teacher model to a simpler student model [41]. In the general approach, the objective function involves minimizing the Kullback–Leibler divergence (KLD) loss between the softened probability distributions of the teacher and student models [42]. A recent study by Kim et al. [43] demonstrated that competitive knowledge distillation from a cumbersome teacher model can be achieved by minimizing the MSE between the logit vectors of the penultimate layer of the teacher and student models, as opposed to using the KLD loss. Inspired by this research, the proposed approach adopts knowledge distillation by minimizing the MSE loss between the output logit vectors of the teacher and student models.

## Methodology

### Overall architecture

Fig 2 illustrates the comprehensive methodology employed in this study. Fig 2A displays the first step of the proposed method. A model is constructed and trained by connecting an

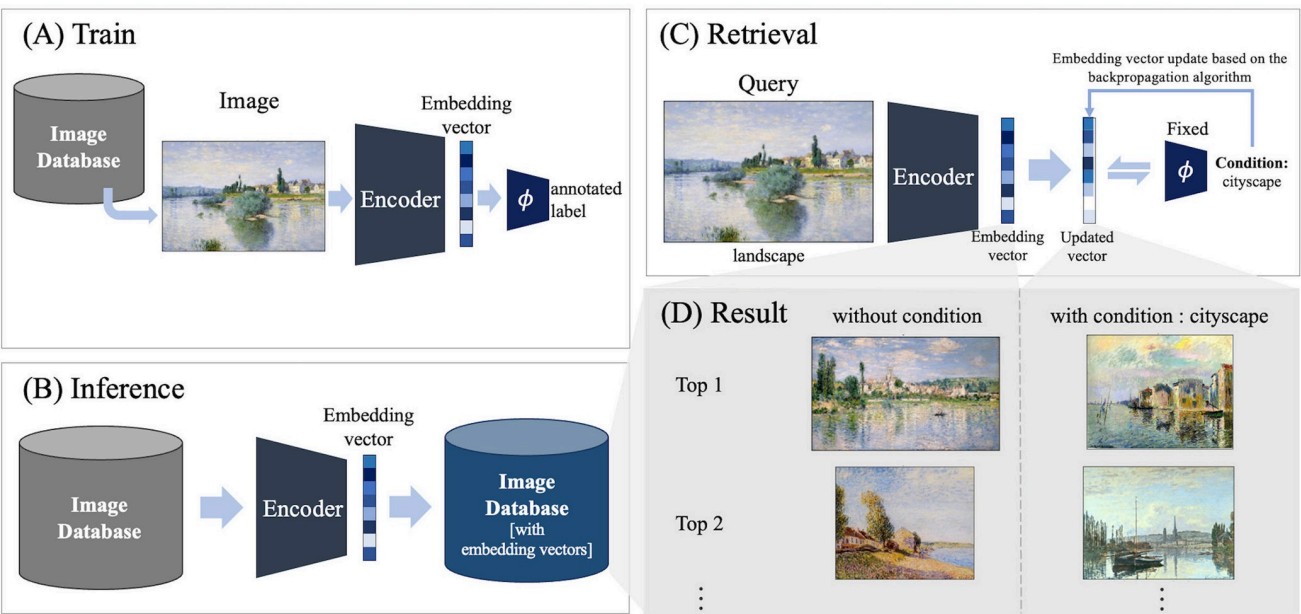

**Fig 2. Proposed conditional image retrieval method.**

encoder and a classification model $\phi$. The encoder extracts the embedding vector of the query image and the model $\phi$ maps the embedding vector to the label.

The encoder model consists of a backbone model and a fully connected (FC) layer dimension reduction layer. The ImageNet pre-trained models VGG16_bn [32] (VGG), Resnet50 [44] (RN), Vision Transformer base [45] (ViT), and ConvNext-base [46] (CN) were used for the backbone model in an ablation study. The embedding vector of the backbone model is extracted from the penultimate layer. Since the vectors from the backbone model are large, a simple FC layer reduces the dimensions of the features from the backbone due to the computational cost and normalizes the embedding size between the backbone models. In all experiments, the backbone model of the encoder is the CN model, which demonstrates the highest performance. and the model $\phi$ consists of only three FC layers. Additionally, each hidden layer in the module $\phi$ is applied with both a batch normalization layer [47] and a dropout layer [48].

Multiple classification models can be applied for the number of properties of interest for the CIR process (e.g., the WikiArt property style and genre classification models are constructed in this study). The general loss function of the proposed models is formulated in Eq 1:

$$\mathcal{L}_{total} = \sum_{i \in N} \alpha_i CE\big(y_i, \ \phi_i(\text{encoder}(x))\big) \tag{1}$$

where $N$ represents the number of classification models, and $\alpha_i$ is the weight score associated with the $i$-th property, reflecting its importance in the overall model. $y_i$ is the ground truth label for the $i$-th property, and $x$ is the input image. The 'encoder'function transforms the input image x into an embedding vector suitable for classification. Each $\phi_i$ is a classification module tailored to the $i$-th property, which processes the encoded image to predict the probability of each class. The cross-entropy loss ($CE$) measures the discrepancy between the ground truth label $y_i$ and the predicted probability output from the classification module $\phi_i$ (encoder(x)).

In the subsequent step shown in Fig 2B, the trained encoder extracts embedding vectors, contributing to the establishment of an embedding database tailored for IR. The embedding database $DB_{emb}$ is constructed as shown in Eq 2:

$$DB_{emb} = \{\, (x,\, z) \,|\, z = encoder(x),\, x \in DB \} \qquad (2)$$

Where x represents a image in the database $DB$ and the encoder is the trained model shown in Fig 2A.

Fig 2C provides a visual representation of the search method. In the context of IR (without conditions), an embedding vector is extracted upon inputting the query image into the encoder. This vector is used to execute a search within the image database generated in Fig 2B, seeking images that demonstrate high similarity to the query vector. Eq 3 represents the process of retrieving the image x that corresponds to the embedding z in the database $DB_{emb}$ with the highest similarity to the query image $x_q$.

$$x^* = \underset{\{x,z\} \in DB_{emb}}{\arg\min}\; \delta\Big(encoder\big(x_q\big), z\Big) \qquad (3)$$

Where $\delta$ is the distance function that measures between the query embedding $z_q$ and z from the database $DB_{emb}$. Cosine similarity is employed in these cases.

In the proposed CIR method, Backward Search, the embedding vector extracted from the encoder is iteratively updated until the model $\phi$ accurately produces the conditioning label. The resulting updated embedding vector is then used to search the embedding vector image database built in Fig 2B, using a process similar to that of IR. More details are provided in the following section.

## Backward search

We propose a Backward Search that utilizes the inductive knowledge of a trained model, as shown in Fig 2A, to perform CIR. The proposed Backward Search finds the optimal embedding that satisfies the conditions while maintaining the features of the query image based on inductive knowledge. Once the optimal embedding is found, it retrieves the closest embedding present in the database, as described in Eq 3, and outputs the corresponding image. An inverse mapping approach is used to find the optimal embedding, and the mathematical expression for obtaining the optimal embedding we propose is shown in Eq 4:

$$z^* = \underset{z}{\arg\min} \left[ \sum_{i=1}^{m} CE\big(y_i^\dagger,\, \phi_i(z)\big) + \lambda R(z - z_0) \right] \qquad (4)$$

where $m$ represents the number of conditions, $z$ is the query embedding, and $z^0$ is the initial latent vector. The condition label is denoted by $y_i^\dagger$, and $\phi_i$ represents the classification module corresponding to the $i$-th condition. CE measures the cross-entropy loss between the specified condition label $y_i^\dagger$ and the predicted probability $\widehat{y}_i$ as $\phi_i(z)$. Additionally, the regularization term $R$ preserves the characteristics of the initial latent vector $z_0$, alleviating the ill-posed problem, and the parameter $\lambda$ balances the contribution of the regularization term.

Specifically, Backward Search finds the optimal $z^*$ by minimizing the cross-entropy loss between $\hat{y}$ and the target condition $y^\dagger$ under the penalty term $R$ as described in Eq 4. Thus, the Backward Search relies on the inductive knowledge of the module $\phi$ in the process of finding the optimal $z^*$. In implementations, As depicted in Fig 2C, we use the iterative approximation suggested in [37], using backpropagation. The proposed method iteratively repeats the mentioned process until z is reaches the embedding vector $z^*$ in the embedding space, which

---

## Algorithm 1 Backward Search

**Require:** Query's embedding $z_q$, target condition $y_i^{\dagger}$, deep neural networks $\phi_i$ classification module tailored to the $i-th$ property, for all condition $i \in \{1, 2, ..., m\}$, weight $\lambda$ of the regularization, and database $DB_{emb}$ with the extracted embedding

**Ensure:** retrieval image $x$ from $DB_emb$

1: Initialize $CE\_loss = 0$; $z_0 = z_q$; **while** $z_q$ not converged **do**
3: **for** $i \in \{1, 2, ..., m\}$ **do**
4: $\hat{y}_i = \phi_i(z_q)$
5: $CE\_loss \mathrel{+}= \text{CrossEntropy}(y_i, \hat{y}_i)$
6: **end for**
7: $loss = CE\_loss + \lambda||z_q - z_0||_1$
8: backpropagation on $z_q$ using the computed $loss$
9: **end while**
10: $x^* = \arg\min_{(x,z)\in DB_{emb}} \delta(z_q, z)$
11: **return** $x^*$

---

**Fig 3. Proposed Backward Search algorithm for conditional image retrieval.**

corresponds to $y^{\dagger}$. The proposed Backward Search algorithm is summarized in Fig 3. Further details on the experimental setup and model implementation are presented in the Implementations section.

### Backward search knowledge distillation

In the proposed method, the number of iterations needs to update the embedding vector until the conditioning label is satisfied results in significant time consumption. In order to reduce the time consumption, we adopt the knowledge distillation method. The KD method consists of a teacher model and a student model, aiming to distill the knowledge from the larger teacher model into the smaller student model so that the student model's outcomes match those of the teacher model. Fig 4 shows the proposed approach to the knowledge distillation method. We consider the updating embedding process based on Backward Search as the teacher model, and we have distilled this knowledge into an autoencoder student model. The autoencoder architecture consists of encoder and decoder modules transferring the input vector concatenated with embedding and condition to the output of the teacher model. In implementing the student model, each module of the student model has only three layers, with a batch normalization layer [47] and a dropout layer [48] applied to the outputs of each hidden layer in the model. A condition is one-hot encoded and concatenated with the query image embedding vector to be input into the student model. The knowledge distillation process is as follows: When z and the condition are provided, the Teacher model produces the updated z using our proposed Backward Search. The Student model feeds the concatenated z with the one-hot encoded condition to predict the updated z as produced by the Teacher model. As

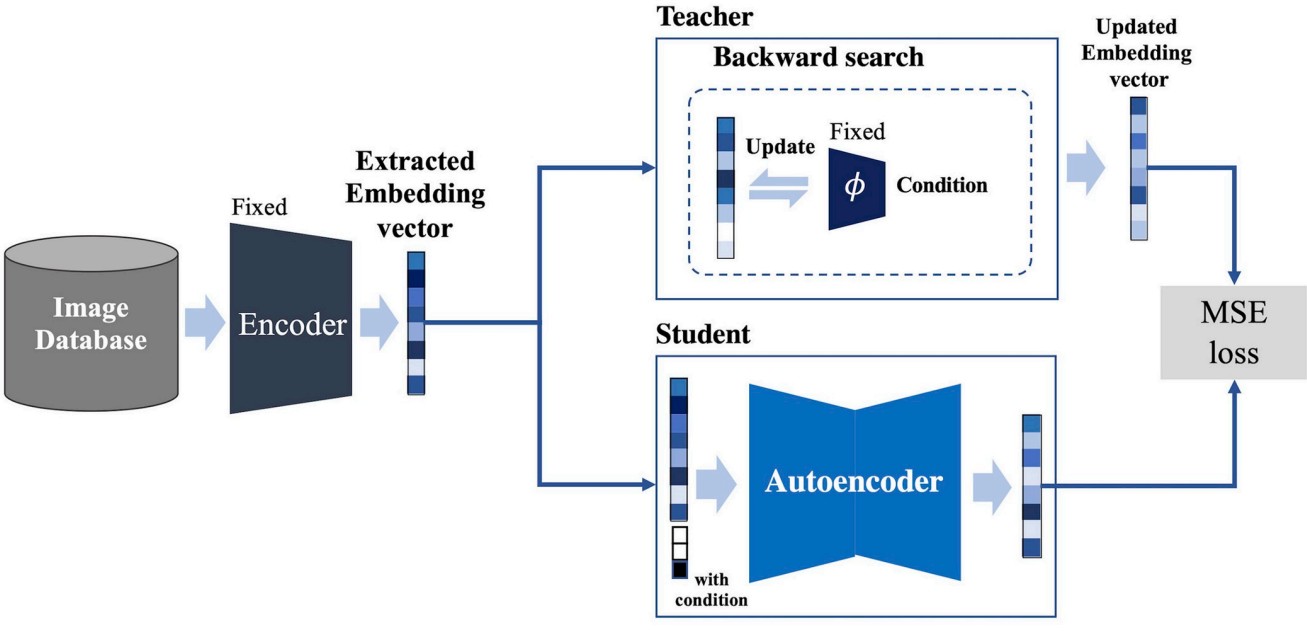

**Fig 4. Proposed knowledge distillation model for the Backward Search.**

demonstrated by Kim et al. [43], we simply minimized the MSE loss as the KD loss between the output logit vectors of the teacher and student models.

**Evaluation metric.**   Jaccard similarity was employed to assess the similarity of the retrieved images with respect to the attributes of the query image. Using the annotated attributions and tags of the images in the dataset, we calculated the Jaccard similarity [49] between the images by matching them. The Jaccard similarity value ranges from 0 to 1, with higher values indicating higher similarity in form and semantic characteristics between the two images. To measure this, we used the average top $k$ mean tag similarity (mTS@k). As the mAP is a widespread metric in IR tasks (2), We employed the mean average precision at $k$ (mAP@k) values by comparing the labels of the retrieved images with the query's conditions.

## Experiments

### Datasets

As Table 1 presents the evaluated benchmark datasets, This study employs the WikiArt [29], Caltech-UCSD Birds-200-2011 (CUBS) [31], and attributes Pascal and Yahoo (aPY) [30] benchmark datasets. The CUBS [31] benchmark is a fine-grained recognition dataset comprising images of 200 distinct bird species, totaling 11,788 images. Each image includes detailed

**Table 1. Datasets and queries used in the experiments.**

| Dataset | Source Dataset | Condition (# classes) | Images | | Query |
|---|---|---|---|---|---|
| | | | **Train** | **Test** | **Test** |
| CUB-bird | CUB [31] | Bird (200) | 8,250 | 3,538 | 1,000 |
| aPY-category | aPY[30] | Subcategory (20) | 6,340 | 6,355 | 1,000 |
| WikiArt-style | WikiArt[29] | Style (27) | 65,155 | 16,289 | 1,000 |
| WikiArt-multi | WikiArt[29] | Style (27), Genre (11) | 65,155 | 16,289 | 1,000 |

annotations, including one subcategory label, 15-part locations, 312 binary attributes, and one bounding box. We randomly divided the CUB dataset into training and testing datasets in a 7:3 ratio.

The aPY benchmark is a coarse-grained dataset comprising 15,339 images from three broad categories, further divided into 32 subcategories. There are only 20 subcategories available in the aPY dataset. Because aPY has multiple objects in an image, we cropped the objects with the bounding box information provided and used the cropped images for training and testing.

WikiArt contains 81,444 works of visual art by various artists, taken from WikiArt.org. Each image has artist, genre, and style labels. In this study, for simplicity, we apply only the style and genre class labels, with 27 style classes and 11 genre classes, including the "unknown genre" class. To assess the CIR performance of the proposed model, we collected attributions for images from the WikiArt dataset [29]. Using Selenium WebDriver, a Python library, we automatically retrieved attributions, such as style, genre, media, and textual descriptions associated with each image from WikiArt.org. Our study calculates the mean tag similarity using not only style and genre but also 141 media types and 5132 unique tags. We enclosed the collected attribution data with the source code.

In this experimental section, we refer to the aPY-category, CUB-bird, WikiArt-style, and WikiArt-multi sets comprising generated image–label pairs as demonstrated in Table 1. Specifically, aPY-category includes 20 subcategories, CUB-bird involves 200 bird species, WikiArt-style includes 27 styles, and WikiArt-multi is a randomly selected style and genre as a condition. Considering the time and cost of model evaluation, this study randomly generated 1000 image–class paired test query sets for each test dataset with a balance between classes. Less than 10 data satisfying an image–class pair based on tags with a Jaccard similarity of 0.1, We also excluded attributions with a high frequency from the CUB datasets based on an occurrence of above 0.8 at the percentile level to achieve clearer results.

During the evaluation process, we assessed our CIR system using test datasets, which included generated image–class pairs such as aPY-category, CUB-bird, WikiArt-style, and WikiArt-mult. We evaluated the system using mAP@10, focusing on the emergence of the conditioning label. Furthermore, we computed the mean tag similarity (mTS@10) between the query and retrieved images.

## Implementations

The experiment was equipped with a Ryzen 9 5900X processor, an RTX3090 GPU, and 64 GB of RAM. The system runs on Ubuntu 22.04 and uses Python 3.9, and Cuda 12.2. The main libraries include Torch 2.0.1, torchvision 0.15, numpy 1.24, sikit-learn 1.3.2, and scipy 1.10. The pre-trained weights for the encoder backbone model were obtained using torchvision.

In the training phase of the model, the edge size of the image was reduced to 224 while maintaining the ratio of image dimensions. Afterward, the images were randomly cropped using the dimensions of 224x224. The cropping tool was RandomResizedCrop, which was implemented in Torchvision with the default options. Each cropped image was randomly flipped horizontally or vertically and normalized with ImageNet images using the mean/sd normalization. The model was trained for 100 epochs. Throughout the training process, the weights of the backbone module were fixed, and the proposed model was fine-tuned using the Adam optimizer [50] with a weight decay rate of 0.0001. The learning rate scheduling method (StepLR), which is implemented in PyTorch, was used with a step size of 30 and a gamma value of 0.1. In the model inference phase, the images were resized in the same manner as they were during the training phase. While testing the model, images were center-cropped without

**Table 2. Results of the Backward Search performance without regularization.**

| Method | CUB-bird | aPY-category | WikiArt-style | WikiArt-multi |
| --- | --- | --- | --- | --- |
| | mAP@10 | mAP@10 | mAP@10 | mAP@10 |
| random search | 0.005 | 0.059 | 0.035 | 0.008 |
| $k$-nearest search | 0.003 | 0.015 | 0.038 | 0.009 |
| Backward Search ($\lambda = 0$) | **0.633** | **0.950** | **0.837** | **0.603** |

using RandomResizedCrop or flipping. Last, inverse mapping was executed with 100 iterations during each trial using a learning rate of 0.2. In addition, early stopping was performed to obtain quicker results during the process. In the experiments, we implemented the t-SNE [8] plot with the default t-SNE option in scikit-learn.

## Backward search without regularization

Table 2 presents the performance of the Backward Search at λ set to zero in Eq (2). In other words, the updated embeddings are not regularized during the Backward Search. In this scenario, the Backward Search updates the the vectors to align with the class condition as much as possible. Under these conditions, the Backward Search achieved mAP@10 scores of 0.633, 0.950, 0.837, and 0.603 for CUB-bird, aPY-category, WikiArt-style, and WikiArt-multi, respectively. Notably, the Backward Search outperformed the random search and $k$-nearest search, indicating that it is effective in the context of CIR tasks.

## Effect of regularization parameter λ

In this study, the regularization penalty term $R$ in Eq (2) was activated to preserve the overall features of the query images and update them to the target condition. In other words, the regularization parameter λ controls the degree of influence of $R$. Fig 5 visualizes the effect of the regularization parameter λ on a sample image. In Fig 5A, the query image has the style and genre characteristics of "Impressionism" and "Portrait," respectively. In the case of this query image, which has the style condition of "Baroque," as presented in Fig 5B, the result of CIR varies accordingly with changes in the value of λ. In ordinary IR ($k$-nearest search), portrait images similar to the query image are retrieved. With the regularization parameter (λ) set to 16, the proposed model retrieves baroque-style portrait images as the top one and two results, exhibiting a high visual similarity to the query image. When λ is four, the top three retrieved images are Baroque-style portrait images. However, the top image is visually less similar, featuring a gray color sketch. With λ set to zero, the proposed model retrieves baroque-style portrait images. Nevertheless, the top one and three images are visually less similar, depicting several people or women. These visualizations illustrate that, as λ decreases, the retrieved images diverge more from the query images, although the retrieved images meet the specified conditions.

Fig 5A depicts a $t$-distributed stochastic neighbor embedding ($t$-SNE) scatterplot of the embedding space for Baroque and Impressionism image embedding vectors. As corresponding to Fig 5b retrieved images, on the t-SNE scatterplot, we marked the query and averaged the embedding vectors of the retrieved images for each λ The scatter plot explicitly illustrates that, as the value of λ decreases, the distance of the embedding space between the query and retrieved images increases. Consequently, a smaller λ allows the retrieval of more images that satisfy the condition, but at the expense of losing the features of the query image. Conversely, with a larger λ, the retrieved images maintain more features of the query image, but this also

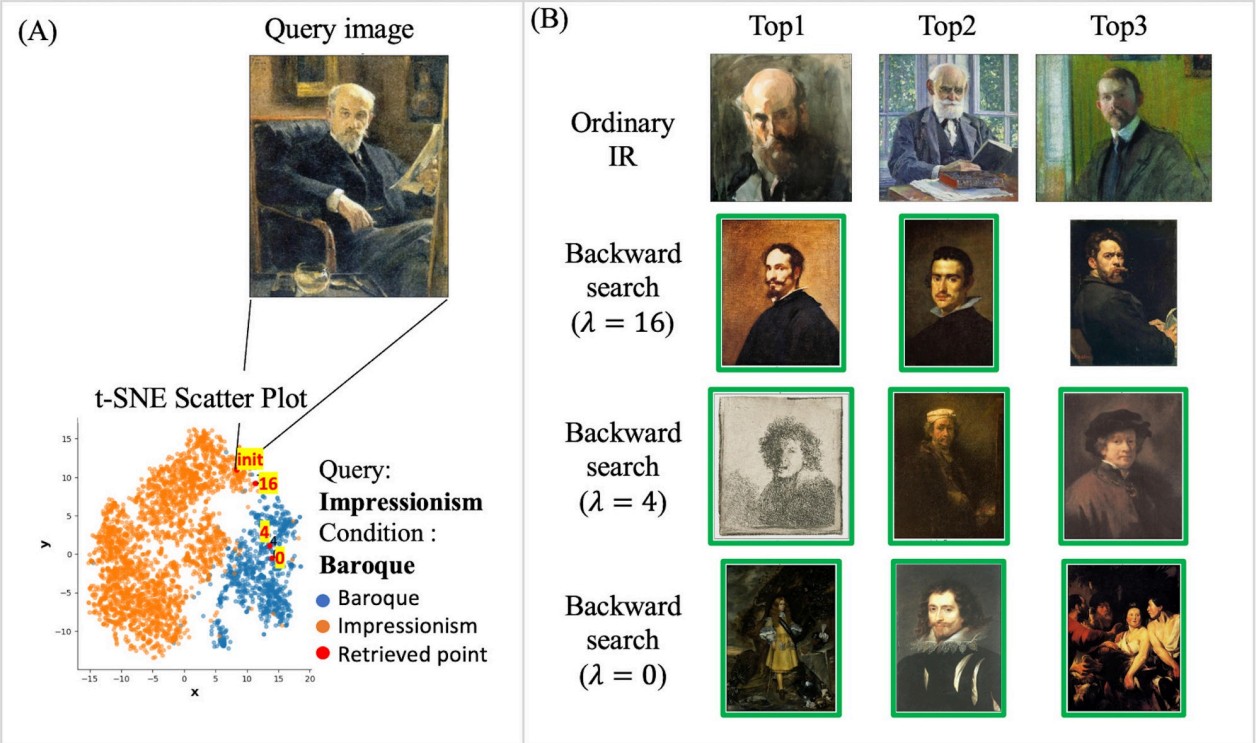

**Fig 5. Backward Search results using various λ regularization parameter settings.** (A) t-SNE scatterplot depicting the embedding space of the Baroque and Impressionism dataset embeddings from WikiArt. The 'init' query image is marked on the plot. (B) Conditional image retrieval (CIR) result for the query image in (A). The embedding vectors of the top three retrieved images are averaged and visualized on the same t-SNE scatterplot in (B). A green box indicates that the retrieved image has the same class as the condition.

leads to fewer images satisfying the condition being retrieved. Therefore, the proposed method allows users to determine the extent to which the features of the query image are reflected in the retrieved images through the λ parameter.

Fig 6 summarizes these trends, showing a plot of the resulting values of mAP@10 and mTS@10 for the proposed model based on λ for each dataset. for all datasets, when λ is large, the retrieved results exhibit high mTS but low mAP. The opposite is true for small λ values. Therefore, the proposed method enables the adjustment of the extent to which the search condition labels and query images reflect the characteristics of the query by selecting an appropriate λ value for the constraints.

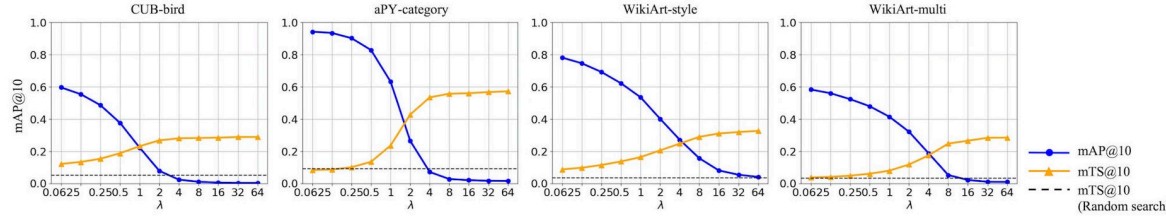

**Fig 6. Variation in performance of mAP@10 and mTS@10 based on the lambda value of the regularization term in the test dataset.**

**Table 3. Experiments involving variations in model architectures and backbone fine-tuning.**

| Back-bone | Fine-tuned | Embedding- size | CUB-bird | aPY-category | WikiArt-style | WikArt-multi | Average |
|---|---|---|---|---|---|---|---|
| | | | mAP@10 | mAP@10 | mAP@10 | mAP@10 | mAP@10 |
| VGG [32] | T | 128 | 0.180 | 0.778 | 0.427 | 0.111 | 0.374 |
| | T | 256 | 0.280 | 0.780 | 0.513 | 0.212 | 0.446 |
| RN [44] | T | 128 | 0.408 | 0.759 | 0.455 | 0.124 | 0.437 |
| | T | 256 | 0.479 | 0.796 | 0.467 | 0.175 | 0.479 |
| ViT [45] | T | 128 | 0.371 | 0.713 | 0.466 | 0.118 | 0.417 |
| | T | 256 | **0.494** | 0.736 | 0.537 | 0.21 | 0.494 |
| CN [46] | | 64 | 0.129 | 0.669 | 0.318 | 0.063 | 0.295 |
| | | 128 | 0.252 | 0.731 | 0.385 | 0.131 | 0.375 |
| | | 256 | 0.330 | 0.748 | 0.46 | 0.167 | 0.426 |
| | T | 64 | 0.192 | 0.751 | 0.428 | 0.096 | 0.367 |
| | T | 128 | 0.407 | 0.779 | 0.520 | 0.186 | 0.473 |
| | T | 256 | 0.481 | **0.800** | **0.595** | **0.289** | **0.541** |

We adjusted the regularization $\lambda$ to maintain an mTS@10 of around 0.150 and then compared the mAP@10 performance. Under these conditions, the average mTS@10 was 0.152, with a standard deviation of 0.0078.

## Ablation study

Table 3 displays ablations performed on various backbone model architectures, investigating the influence of backbone fine-tuning on the CUB-bird, aPY-category, WikiArt-style, and WikiArt-multi datasets. In order to facilitate a consistent comparison of the mean Average Precision (mAP) across models, we adjusted the regularization parameter $\lambda$ to fix the mTS@10 at a value of 0.15. This allowed for a controlled evaluation of mAP, ensuring that the comparisons were made under uniform threshold settings, as outlined in Table 3. Larger embeddings demonstrate improved performance, and the models with fine-tuned backbones outperform those without fine-tuning. Despite the anticipated performance improvement with an increasing embedding size, we capped the size at 256 due to the computational constraints. Consequently, the experiments favor the ConvNext-base model with a 256-dimensional embedding and a fine-tuned backbone during the training process as the optimal configuration.

## Comparison with CoIR models

Comparison experiments were conducted using the CoIR model. we employed zero-shot models such as CoVR (21), Pic2Word (19), and SEARLE (20) along with a conventional CoIR model, CLIP4CIR (18). The CoVR and CLIP4CIR models used relational triplets, while the Pic2Word (19) and SEARLE (20) models were based on image-caption pairs. The specifics are as follows: CoVR was trained on the WebVid-CoVR dataset, Pic2Word on the CC [51] dataset, and SEARLE on the ImageNet1K [52] dataset. For CLIP4CIR, we used a model trained on the FashionIQ [27] dataset. We utilized the pretrained weights provided by the respective repositories. During the experiments, the CoIR's reference image and conditioning text were derived from the query image and conditioning label generated for this study, described in the Datasets section. The retrieved results were then evaluated using the same method as the Backward Search.

Table 4 presents the outcomes of this comparative study on the CUB-bird, aPY-category, WikiArt-style, and WikiArt-multi datasets. Although the Backward Search simply learns image and image-level labels relations, our proposed model surpassed the CoIR models in

**Table 4. Comparison study between the proposed method and state-of-the-art CoIR models on the CUB, aPY, and WikiArt datasets.**

| Model | CUB-bird | | aPY-category | | WikiArt-style | | WikiArt-multi | | Average | |
|---|---|---|---|---|---|---|---|---|---|---|
| | mAP @10 | mTS @10 | mAP @10 | mTS @10 | mAP @10 | mTS @10 | mAP @10 | mTS @10 | mAP @10 | mTS @10 |
| random search | 0.005 | 0.051 | 0.059 | 0.089 | 0.035 | 0.036 | 0.008 | 0.033 | 0.027 | 0.052 |
| SEARLE [24] | 0.373 | 0.062 | 0.263 | 0.087 | 0.166 | 0.017 | 0.085 | 0.018 | 0.222 | 0.046 |
| Pic2Word [23] | 0.311 | 0.071 | 0.651 | 0.125 | 0.173 | 0.026 | 0.109 | 0.020 | 0.311 | 0.061 |
| CoVR [25] | 0.263 | 0.070 | **0.868** | 0.147 | 0.196 | 0.024 | 0.004 | 0.025 | 0.333 | 0.067 |
| CLIP4CIR [22] | 0.357 | 0.070 | 0.778 | 0.129 | 0.332 | 0.035 | 0.145 | 0.038 | 0.403 | 0.068 |
| Backward Search | **0.481** | **0.149** | 0.800 | **0.147** | **0.595** | **0.152** | **0.289** | **0.147** | **0.541** | **0.149** |

The results of the SEARLE, Pic2Word, CoVR, and CLIP4CIR models used in the comparative study are not the originally reported results from their respective research. We conducted experiments on the datasets to compare with the results of the proposed Backward Search method.

mAP@10 performance on the CUB-bird, WikiArt-style, and WikiArt-multi datasets and achieved the second-highest score on the aPY-category dataset. Despite solely learning categorical boundaries, the tag similarity of our proposed model outperforms other models across all datasets, indicating the successful preservation of query image features using the Backward Search. This outcome highlights the substantial benefit that the proposed method effectively performs CIR by training the classification task.

Additionally, the performance of WikiArt-multi, which involves satisfying two conditions, is nearly halved compared to WikiArt-style which has only one condition. This highlights the challenge of retrieving images that meet multiple conditions. Except for the SEARLE, The aPY-category data exhibited relatively high performance across all models, likely due to the clear distinction between class data and class names formed by common vocabulary. Specifically, the SEARLE model shows lower performance on average compared to other models. This is likely because the SEARLE model is designed to maintain image-level concepts and transform contextual information. The concept at this time is mostly the main instance present in the image. Our proposed model, which maintains the contextual information of the image and performs CIR based on human-defined image-level concepts, differs from the SEARLE model that transforms contextual information. As a result, our proposed model seems to perform better in experiments involving concept transformation. Notably, the relational triplet-based CLIP4CIR [22] and CoVR models displayed higher and more stable average performance than the image-text-based models, suggesting that triplet-based models may have an advantage in learning general relations within datasets.

Figs 7 and 8. depict the qualitative results of the models for each test dataset. Fig 7 displays retrieval results for the WikiArt dataset using the WikiArt-style test query. The query of the "Art Nouveau Modern" image has the condition style of "Ealy Renaissance". The proposed method retrieves all "Ealy Renaissance" paintings except for the top 2 positioned images. In contrast, in the comparison model, most images either do not satisfy the condition label.

Fig 8 illustrates the retrieval results for the WikiArt dataset using the WikiArt-multi test query. The query image, depicting a two-person in the style of painting in Rococo with sketch, has the conditions of Northern Renaissance and portrait. The proposed model searches for three pictures with people that satisfy both conditions. Even the other images were retrieved while maintaining the context of the query image. In contrast, the comparison model may retrieve images satisfying one of the conditions or featuring people and three images satisfy both conditions but the lost sketch texture information. These qualitative analyses demonstrate that the proposed model effectively retrieves images corresponding to specific condition labels during the search process while preserving the characteristics of the query image.

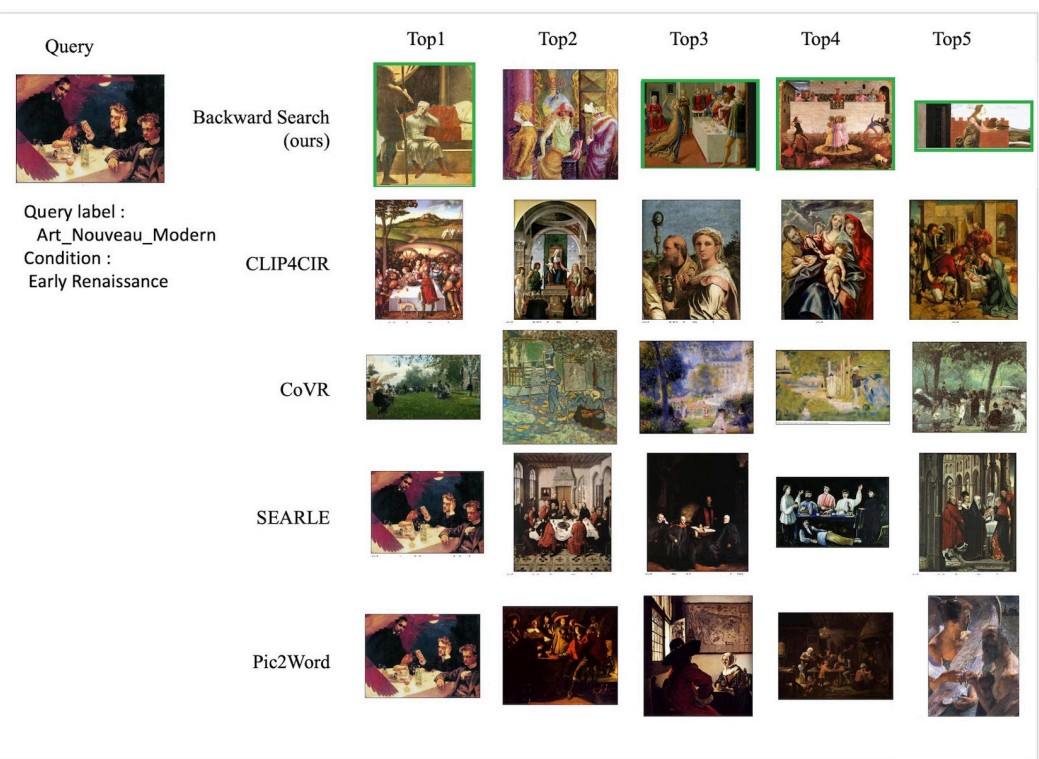

**Fig 7. Qualitative result of WikiArt test dataset with WikiArt-style query.** We mark correct retrieved images with green boxes for the best view.

## Number of iterations

The proposed Backward Search method updates the embedding vector through iteration, which requires a certain execution time. To measure the time required in each dataset, the Backward Search recorded the number of iterations and the time taken. Table 5 lists the results, detailing the number of updates and the time required for the method to fulfill the condition for a single query in each dataset. Notably, as the embedding size increases, the number of iterations increases, and processing time lengthens. in our experimental hardware setup (see section Implementations), It takes an average of 12, 16, and 16 ms, for 128 and 256 embeddings to process one query, with the per-query processing speed expected to decrease further when batch processing is implemented.

## Knowledge distillation results

To alleviate the time consumption of the Backward Search approach, we employed the proposed method as a teacher model to distill the backward knowledge to a student model, as detailed in the Backward Search knowledge distillation section. The results of this knowledge distillation are presented in Table 6. The student model processes queries approximately 160 times faster than the Backward Search method. However, it shows a decrease of approximately 15% in mAP@10 and an increase of about 13% in mTS@10, compared to the teacher model. When comparing the retrieved images of both models, the similarity of the retrieved images suggests that the two models produce relatively comparable results. For instance, the aPY dataset demonstrates a Jaccard similarity of 0.477, indicating that, on average, about 6 out of 10

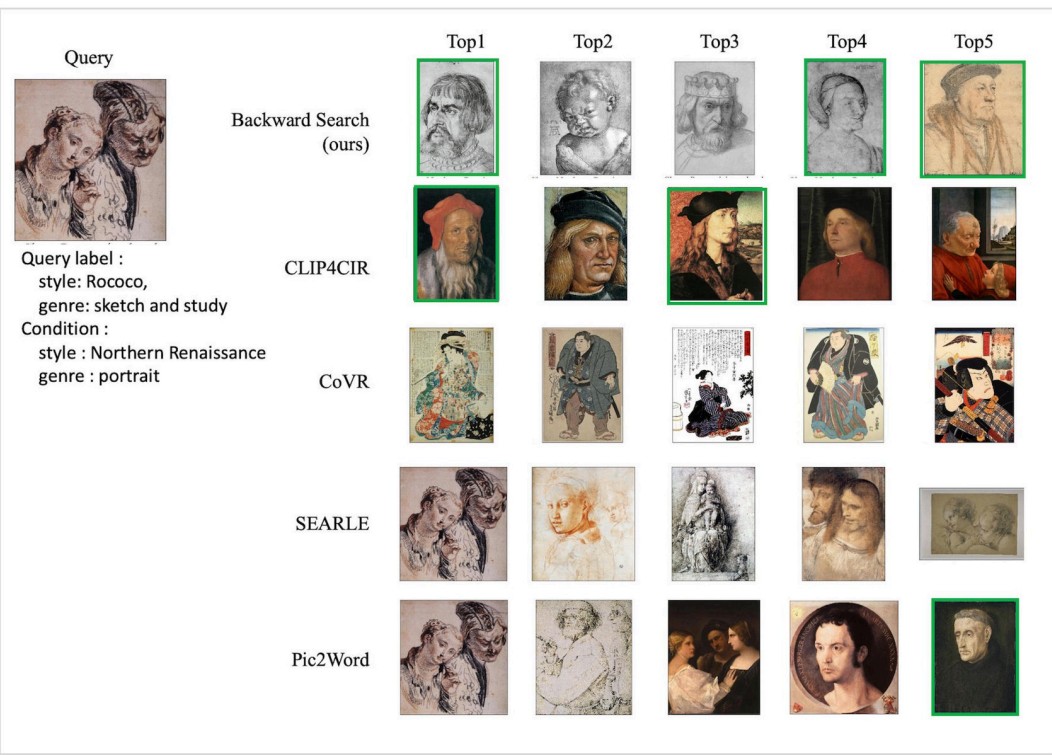

**Fig 8. Qualitative result of WikiArt test dataset with WikiArt-multi query.** We mark correct retrieved images with green boxes for the best view.

**Table 5. Iterations and time consumption per query for the Backward Search.**

| Dataset | Embedding size | Backward Search | | |
|---|---|---|---|---|
| | | Iteration mean | Iteration Std. | Processing time |
| CUB-bird | 128 | 25.01 | 3.45 | 13.05 ms |
| | 256 | 33.63 | 3.77 | 18.22 ms |
| aPY-category | 128 | 22.97 | 4.74 | 11.75 ms |
| | 256 | 28.95 | 5.97 | 15.96 ms |
| WikiArt-style | 128 | 20.79 | 4.65 | 10.56 ms |
| | 256 | 26.49 | 5.63 | 14.65 ms |
| WikiArt-multi | 128 | 17.50 | 3.32 | 13.55 ms |
| | 256 | 17.97 | 3.18 | 14.80 ms |

**Table 6. Results of knowledge distillation of the Backward Search.**

| Dataset | Teacher (Backward Search) | | | Student | | | Retrieved similarity |
|---|---|---|---|---|---|---|---|
| | mAP@10 | mTS@10 | Processing time | mAP@10 | mTS@10 | Processing time | |
| CUB-bird | 0.481 | 0.149 | 18.22 ms | 0.344 | 0.199 | 0.098 ms | 0.616 |
| aPY-category | 0.800 | 0.147 | 15.96 ms | 0.794 | 0.146 | 0.102 ms | 0.477 |
| WikiArt-style | 0.595 | 0.152 | 14.65 ms | 0.508 | 0.168 | 0.097 ms | 0.700 |
| WikiArt-multi | 0.289 | 0.147 | 14.80 ms | 0.242 | 0.160 | 0.101 ms | 0.540 |

The retrieved similarity was calculated by averaging the Jaccard similarity of the top 10 images searched by the teacher and student model for a single query.

retrieved images are matched. Thus, despite its relative simplicity, the student model can effectively distill the knowledge of the Backward Search.

## Multi-conditions image retrieval

The proposed CIR method also operates with multiple conditions, producing an image that fulfills all conditions. With various specified conditions, the model searches for more diverse types of images. Fig 9 depicts visualized multiple conditions retrieval results. The query image has the style "Post Impressionism" the genre of "sketch and study" and depicts a countryside village and people. The retrieved images in Fig 9 maintain the genre of the query image. Furthermore, without conditions, only the top three positioned image were retrieved from "Post Impressionism" style paintings. When the style "Post Impressionism" was given as a condition, the style of "Post Impressionism" images was returned by retaining the genre of "sketch and study" of the query image. Two Conditions introduces multiple conditions, such as the style of "Post Impressionism" the and genre of "landscape", and all images belonging to this genre and style are retrieved. All the images have the style of Post Impressionism but are retrieved as colored landscapes. Last, when the "sketch and study" genre condition is added to the previous conditions, Interestingly, the images that correspond to all conditions are detected. In this manner, the proposed method enables the retrieval of images tailored to the user's intentions by assigning various conditions and It demonstrates that our proposed model can effectively retrieve large image datasets with complex contextual information.

## Limitations

The proposed Backward Search approach leverages the inductive knowledge acquired by the trained model, which necessitates supervised label learning to be employed as conditions,

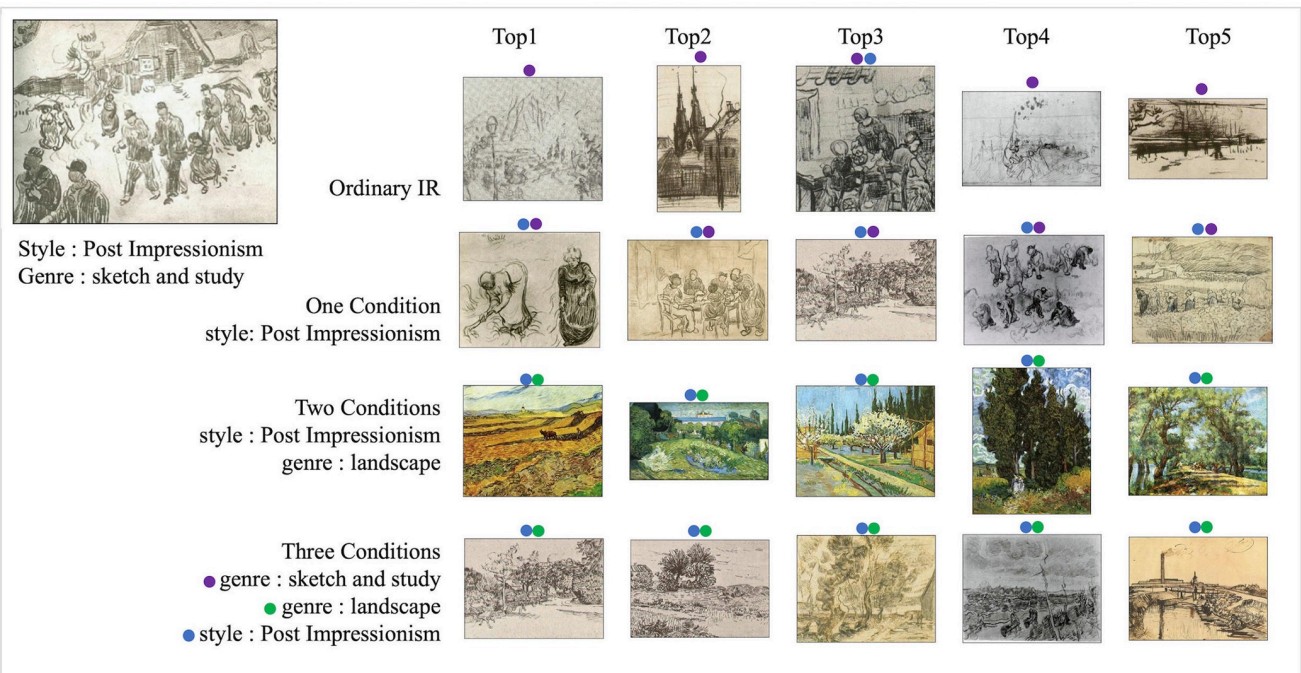

**Fig 9. Qualitative multiple condition image retrieval results.** We marked the labels of the WikiArt dataset on each retrieved image with colored circles. purple is "sketch and study", green is "landscape", and blue is "Post Impressionism".

requiring a substantial amount of data for generalized performance. Furthermore, the Backward Search process requires iterative updates to find the optimal embedding vector, which is time-consuming. To address these issues, we employ knowledge distillation successfully reducing the computing time, as detailed in Table 6. However, the mAP retrieval performance of the student model is observed to slightly decrease.

The proposed method conducts CIR at the image-level label, presenting limitations compared to models such as CoIR which accept natural language text as conditions. The FashionIQ and CIRR datasets used in the CoIR models are labeled with the relation between images and natural language, making it challenging to experiment with our proposed model. Therefore, in this study, we conducted comparative experiments using aPY, CUB, and Wikiart datasets. In future work, we plan to extend our proposed model to enable conditional search with natural language as well, and to experiment with the FashionIQ and CIRR datasets. Additionally, in the comparison experiments with CoIR, the divergent vocabulary coverage and domain shift in the comparative model may cause the performance decline. This suggests that the ZS-CoIR model could also experience degradation in performance when dealing with domain-specific datasets. However, As CoIR methods require predefined relational triplets or many well-described captions, our approach does not require such structures and can perform conditional searches solely based on image-level label datasets which have many public datasets. Additionally, future research could explore incorporating the CLIP model to study inversion-based CIR models that use text as conditions, which could mitigate the limitations of the proposed method.

## Conclusions

This study proposes a conditional image retrieval based on Backward Search with inductive knowledge. The proposed approach enables searches that are aligned with user intent by utilizing image-level labels as conditions. By employing the aPY, CUB, and WikiArt datasets, the performance of the proposed method was evaluated both qualitatively and quantitatively. Comparative evaluations reveal that the proposed model has a competitive performance compared to the CoIR model. While the proposed method has limitations in performing conditional retrieval only at the image-level label compared to the CoIR method, it offers significant advantages. Specifically, it effectively preserves and controls the query image's contextual information without the need for triplets or detailed image captions. However, The Backward Search operates iteratively and can be time-consuming depending on the environment. To address these drawbacks, we implement a knowledge distillation method that streamline the Backward Search process. As a result, the proposed student model successfully reduces the computing time with only a slight decrease in retrieval performance. For future work, we intend to explore the application of the Backward Search method for conditioning with natural language text, utilizing CLIP models. We believe the CLIP model, which constructs an integrated embedding space of multi-modalities, will enable our proposed Backward Search to retrieve images based on natural language conditions.

## Author Contributions

**Conceptualization:** Donghwan Lee, Wooju Kim.

**Data curation:** Donghwan Lee.

**Formal analysis:** Donghwan Lee.

**Funding acquisition:** Wooju Kim.

**Investigation:** Donghwan Lee.

**Methodology:** Donghwan Lee.

**Project administration:** Donghwan Lee, Wooju Kim.

**Resources:** Wooju Kim.

**Software:** Donghwan Lee.

**Supervision:** Wooju Kim.

**Validation:** Donghwan Lee, Wooju Kim.

**Visualization:** Donghwan Lee.

**Writing – original draft:** Donghwan Lee.

**Writing – review & editing:** Donghwan Lee, Wooju Kim.

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
