## [Decision Letter · Decision Letter 0]

15 May 2024

PONE-D-24-16658Backward Inductive Deep Neural Image SearchPLOS ONE

Dear Dr. Kim,

Thank you for submitting your manuscript to PLOS ONE. After careful consideration, we feel that it has merit but does not fully meet PLOS ONE’s publication criteria as it currently stands. Therefore, we invite you to submit a revised version of the manuscript that addresses the points raised during the review process.

We look forward to receiving your revised manuscript.

Kind regards,

Dr. S. M. Anas, Ph.D.(Structural Engg.), M.Tech(Earthquake Engg.)

Academic Editor

PLOS ONE

Journal Requirements:

"NO authors have competing interests".

4. We note that Figures 1, 2, 4, 6, 7, 8 9 and 10 in your submission contain copyrighted images. All PLOS content is published under the Creative Commons Attribution License (CC BY 4.0), which means that the manuscript, images, and Supporting Information files will be freely available online, and any third party is permitted to access, download, copy, distribute, and use these materials in any way, even commercially, with proper attribution. For more information, see our copyright guidelines: http://journals.plos.org/plosone/s/licenses-and-copyright.

       1. You may seek permission from the original copyright holder of Figure(s) [#] to publish the content specifically under the CC BY 4.0 license.  

Additional Editor Comments :

Dear Authors,

I hope this email finds you well.

I am writing to inform you about the status of your manuscript entitled "Backward Inductive Deep Neural Image Search" [PONE-D-24-16658], which has undergone peer review. Both reviewers have provided comprehensive feedback, and unfortunately, they have both recommended Major Revision based on serious comments related to the layout and content of the manuscript.

Upon conducting a preliminary assessment of the manuscript and carefully reviewing the comments provided by the reviewers, I concur with their assessment that major revisions are necessary to address the concerns raised. Therefore, I have decided to request Major Revision for your manuscript, subject to the approval of the editorial board.

The reviewers' comments highlight significant issues that need to be addressed to improve the quality and clarity of your manuscript.

I kindly request you to submit the revised manuscript along with a detailed response to each of the reviewers' comments through the submission system by the specified deadline. If you require any clarification or assistance during the revision process, please do not hesitate to contact me.

Important note from Academic Editor, Dr. S. M. Anas: -

I would like to bring to your attention that citing the papers suggested by the reviewers is not mandatory for your revised manuscript. It is entirely up to you whether or not you choose to include the suggested papers in your revised version. The reviewers have provided these suggestions to enhance the quality and credibility of your research, but ultimately, the decision is yours. You have the freedom to decline including any of the suggested papers in your revised manuscript if you feel they are not relevant or do not add value to your study.

Thank you for your attention to this matter, and I look forward to receiving your revised manuscript.

Best regards,

Dr. S. M. Anas

Academic Editor

PLOS ONE

Reviewers' comments:

Reviewer's Responses to Questions

**Comments to the Author**

1. Is the manuscript technically sound, and do the data support the conclusions?

Reviewer #1: Partly

Reviewer #2: Yes

2. Has the statistical analysis been performed appropriately and rigorously? 

Reviewer #1: No

Reviewer #2: Yes

3. Have the authors made all data underlying the findings in their manuscript fully available?

Reviewer #1: Yes

Reviewer #2: Yes

4. Is the manuscript presented in an intelligible fashion and written in standard English?

Reviewer #1: Yes

Reviewer #2: Yes

5. Review Comments to the Author

Reviewer #1: The following points need to be addressed in the revision:

1. The title and abstract are not congruent with each other.

2. The abstract is not adequate in length and structure. Motivation needs more focus.

3. The main innovation and contribution of this research should be clarified in the abstract and introduction.

4. The salient results of the work should be mentioned in the abstract along with the performance metrics used in the work.

5. In the contributions mentioned, nothing is talked about the 4 trillion pages from which you intend to search. Make a comprehensive reply to this point in your list of contributions.

6. Add the recent reference in related work: https://doi.org/10.3390/app12104943

7. Thoroughly check your equations and the variables to remove the errors.

8. Page 15, Line 210: the tables should follow the text. Correct this problem.

9. Shift the implementation details to the beginning of the Results.

10. The word inductive was used in the beginning and then at the end in the Limitations, and also in the title. You need to explain in the article why you are using it. It is a by-default feature of learning.

11. The conclusions section is more than expanded. It needs to be summarized.

Reviewer #2: In this article, the authors introduce a novel image retrieval approach titled "Backward Inductive Deep Neural Image Search". The authors are requested to thoroughly address the raised questions and undertake necessary responses or revisions.

1- In the first paragraph of the Introduction section, the authors have referred to handcrafted feature extraction methods. The authors could also discuss the advantages of these approaches compared to neural network-based approaches, referencing [R1].

[R1] (2024). Retrieving images with missing regions by fusion of content and semantic features. Multimedia Tools and Applications, 1-23. https://doi.org/10.1007/s11042-024-18370-1

2- In the second paragraph of the Introduction section, the authors mention the extraction of low-level features from deeper layers of neural networks. However, it should be noted that low-level features are extracted from early layers and semantic features from deeper layers of neural networks. References [R2-R3] can be explored and cited for this purpose.

[R2] (2023). Content-based image retrieval using handcraft feature fusion in semantic pyramid. International Journal of Multimedia Information Retrieval, 12(2), 21. https://doi.org/10.1007/s13735-023-00292-7

[R3]. (2023). Efficient deep feature based semantic image retrieval. Neural Processing Letters, 55(3), 2225-2248. https://doi.org/10.1007/s11063-022-11079-y

3- To provide further clarity, it is suggested to present the proposed approach algorithmically step by step.

4- The comparison of the results of the proposed approach with other similar approaches is reported in Table 4 across four datasets. Considering that no results have been reported on similar datasets in any of the references [18-21], it is necessary to elaborate on the reported results in Table 4.

5- It is recommended to evaluate the results of the proposed approach on FashionIQ and CIRR datasets and compare these results with those reported in references [18-21].

6- More explanation is needed regarding the innovation of this approach compared to reference [20].

7- In the Methodology section, the use of knowledge distillation requires further discussion and presentation of more detailed specifics.

8- Details regarding the training of the model have not been mentioned.

6. PLOS authors have the option to publish the peer review history of their article (what does this mean?). If published, this will include your full peer review and any attached files.

Reviewer #1: **Yes: **Prof Dr Shahzad Ahmad Qureshi

Reviewer #2: No

---

## [Author Response · Author response to Decision Letter 0]

28 Jun 2024

Response to Reviewer’s Comments

The authors express their appreciation to the editor, academic editor, and reviewers for their insightful comments and recommendations, which have enhanced the quality of our manuscript. The manuscript has been updated based on these comments. Below, we list the detailed changes made to the manuscript and our responses to the questions, using "C" to denote Comment and "R" for Response.

To Academic Editor 

C1: Please ensure that your manuscript meets PLOS ONE's style requirements, including those for file naming. The PLOS ONE style templates can be found at link1 and link2

R1: We employed the PLOS template to format our manuscript and carefully verified its compliance with the PLOS ONE style guidelines, ensuring it meets all requirements. Should any errors remain, please feel free to contact us. Thank you. 

C2: Please note that PLOS ONE has specific guidelines on code sharing for submissions in which author-generated code underpins the findings in the manuscript. In these cases, we expect all author-generated code to be made available without restrictions upon publication of the work. Please review our guidelines at the link and ensure that your code is shared in a way that follows best practices and facilitates reproducibility and reuse. 

R2: We make available the code developed in this study without restrictions under the PLOS ONE guidelines on code sharing. The code is available on GitHub at the following URL: https://github.com/dhlee-work/BackwardSearch. A mention of this disclosure is included in the last line of the Abstract.

C3: Thank you for stating the following in your Competing Interests section: 

"NO authors have competing interests". Please complete your Competing Interests on the online submission form to state any Competing Interests. If you have no competing interests, please state "The authors have declared that no competing interests exist.", as detailed online in our guide for authors at the link This information should be included in your cover letter; we will change the online submission form on your behalf.

R3: We apologize for the incomplete response regarding our Competing Interests section. The authors have declared that no competing interests exist.

C4: We note that Figures 1, 2, 4, 6, 7, 8 9, and 10 in your submission contain copyrighted images. All PLOS content is published under the Creative Commons Attribution License (CC BY 4.0), which means that the manuscript, images, and Supporting Information files will be freely available online, and any third party is permitted to access, download, copy, distribute, and use these materials in any way, even commercially, with proper attribution. For more information, see our copyright guidelines: link. We require you to either (1) present written permission from the copyright holder to publish these figures specifically under the CC BY 4.0 license, or (2) remove the figures from your submission: 

R3: We apologize for our negligence in correctly utilizing copyrighted images. We have made every effort to address the copyright issue. In our search for information on United States copyright law, we discovered the following details: All works first published or released in the United States before January 1, 1929, have lost their copyright protection 95 years later, effective January 1, 2024. In the same manner, works published in 1929 will enter the public domain as of January 1, 2025, and this cycle will repeat until works published in 1977 enter the public domain on January 1, 2073.

We have revised or removed figures in our paper in line with copyright law. As a result, Figures 1, 2, 5, 9, 10, and 11 were revised and Figures 7 and 8 were removed.

1) For the wikiart dataset, we visualized search results using paintings published before 1929 that are in the public domain, ensuring these changes did not compromise the consistency of the content originally presented in the submitted manuscript. The list of names of paintings used has been added to the released code.

2) Due to potential copyright restrictions, we removed figures from the CUB and aPY dataset from our paper. However, if you are interested in receiving visualization results for these datasets, please contact the authors, and we will provide them.

 

To Reviewer 1

C1: The title and abstract are not congruent with each other. 

R1: We understand the concerns raised in the comment, which appear to stem from an inadequate explanation of the abstract before revision. In the revised abstract from Page 2, Line 22, we clarify the description of our proposed method, which uses an inverse mapping approach to conditional image retrieval based on the inductive knowledge of the trained neural network model. Thus, we titled our study "Backward Inductive Deep Neural Image Search."

C2: The abstract is not adequate in length and structure. Motivation needs more focus. 

R2: Thank you for advising us on the inappropriate length and structure of the abstract. We revised the abstract to focus more on the motivation and improved its structure by presenting the contributions of the research and specific results.

C3: The main innovation and contribution of this research should be clarified in the abstract and introduction. 

R3: We appreciate your advising us on the importance of clearly stating the main innovation and contribution in the abstract and introduction. We have revised both the abstract and the introduction to clarify the main innovation and contribution more effectively. 

The sentence describing innovation is as follows:

Abstract: Page 2, Line 22

- In this work, we show that by exploring the inductive knowledge of a model, CIR at the image-level concept can be achieved by using an inverse mapping approach.

Introduction: Page 4, Line 70

- In this study, we demonstrate that CIR at the image label can be achieved through an inverse mapping approach by leveraging the model’s inductive knowledge while maintaining the query's context. The sentence describing 

Contributions is as follows:

Abstract: Page 2, Line 28

- we introduce the Backward Search method that enables single and multi-conditional image retrieval. Moreover, we efficiently reduce the computation time by distilling the knowledge.

Introduction: Page 4, Line 87, Line 106

- Based on the method described above, we introduce a Backward Search method at the image-level labels that enables both single and multi-conditional image retrieval while preserving the contextual information...

- The contributions of this study are as follows: (1) We propose a conditional image retrieval method, Backward Search, using a model only trained only on a pair of images and image-level labels by leveraging inductive knowledge. (2) The proposed …

C4: The salient results of the work should be mentioned in the abstract along with the performance metrics used in the work. 

R4: Thank you for the suggestion to include specific performance details of the model in the abstract. We have added this information towards the end of the abstract. The details can be found on Page 2, Lines 31-34.

C5: In the contributions mentioned, nothing is talked about the 4 trillion pages from which you intend to search. Make a comprehensive reply to this point in your list of contributions. 

R5: Thank you for advising us to further elaborate on the contributions of our proposed methodology. Following your advice, we have highlighted that our methodology enables efficient searching within large image datasets with complex textures. This contribution has been detailed on Page 4, Line 93, and Page 5, Line 111.

C6: Add the recent reference in related work: https://doi.org/10.3390/app12104943

R6: Thank you for providing the additional references necessary for our study. We included this paper as the reference for our research. This information can be found on Page 5, Line 124.

C7: Thoroughly check your equations and the variables to remove the errors.

R7: We thoroughly checked the equations and variables to remove any errors, and extensively revised the equations and their descriptions to correctly explain the proposed method. If you find any incorrect equations or variables, please inform the authors, and we will verify and correct them.

C8: Page 15, Line 210: the tables should follow the text. Correct this problem. 

R8: Thank you for your specific advice regarding the format of our paper. We have revised it so that all tables now follow the text.

C9: Shift the implementation details to the beginning of the Results.

R9: Thank you for your specific advice regarding the readability of our paper. We moved the implementation section to the beginning of the Results.

C10: The word inductive was used in the beginning and then at the end in the Limitations, and also in the title. You need to explain in the article why you are using it. It is a by-default feature of learning.

R10: As you mentioned, although "inductive" is a by-default feature of learning, we have used the term "inductive" to describe our method that performs conditional searching in a backward manner based on the inductive knowledge of the trained model. This information is noted on Page 2, Line 23, and Page 4, Line 71.

C11: The conclusions section is more than expanded. It needs to be summarized.

R11: Thank you for your advice on the format of the conclusion section. We summarized the conclusion section into a single paragraph.

To Reviewer 2

C1: In the first paragraph of the Introduction section, the authors have referred to handcrafted feature extraction methods. The authors could also discuss the advantages of these approaches compared to neural network-based approaches, referencing [R1].

[R1] (2024). Retrieving images with missing regions by fusion of content and semantic features. Multimedia Tools and Applications, 1-23. https://doi.org/10.1007/s11042-024-18370-1

R1: Thank you for recommending a reference for our paper. We cited R2 (https://doi.org/10.1007/s13735-023-00292-7) to explain the advantages of handcrafted feature extraction approaches. This reference is cited on Page 5, Line 124

C2: In the second paragraph of the Introduction section, the authors mention the extraction of low-level features from deeper layers of neural networks. However, it should be noted that low-level features are extracted from early layers and semantic features from deeper layers of neural networks. References [R2-R3] can be explored and cited for this purpose.

[R2] (2023). Content-based image retrieval using handcraft feature fusion in semantic pyramid. International Journal of Multimedia Information Retrieval, 12(2), 21. https://doi.org/10.1007/s13735-023-00292-7

[R3]. (2023). Efficient deep feature based semantic image retrieval. Neural Processing Letters, 55(3), 2225-2248. https://doi.org/10.1007/s11063-022-11079-y

R2: Thank you for recommending references for our paper. We cited R1 (https://doi.org/10.1007/s11042-024-18370-1) and R3 (https://doi.org/10.1007/s11063-022-11079-y) to describe the features extracted at different depths of neural networks. These references are cited on Page 3, Line 49.

C3: To provide further clarity, it is suggested to present the proposed approach algorithmically step by step.

R3: Thank you for your advice on describing the algorithm proposed in our paper. We structured the description of the Overall architecture section step by step, and also represented the proposed Backward Search with pseudocode algorithm, as shown in Fig3.

C4: The comparison of the results of the proposed approach with other similar approaches is reported in Table 4 across four datasets. Considering that no results have been reported on similar datasets in any of the references [18-21], it is necessary to elaborate on the reported results in Table 4.

R4: Thank you for your advice on accurately describing the comparative experimental results. We specifically detailed in Table 4 that the experimental results of the comparison models are from experiments defined for this study.

C5: It is recommended to evaluate the results of the proposed approach on FashionIQ and CIRR datasets and compare these results with those reported in references [18-21]. 

R5: Thank you for your advice on conducting additional data experiments for this paper. Unfortunately, the FashionIQ and CIRR datasets, which annotate the relationship between images and natural language, are not suitable for experimenting with the model proposed in this study. In future work, we will aim to enable conditional search based on natural language, which can be evaluated with the FashionIQ and CIRR datasets. This response is noted on the Limitation section Page 20, Lines 512-512

C6: More explanation is needed regarding the innovation of this approach compared to reference [20

R6: Thank you for advising us to provide a detailed comparison between the proposed model and reference [20]. We have added further explanation comparing our model with the SEARL model.

- The SEARLE is a zero-shot model designed to maintain image-level concepts while transforming contextual information. The SEARLE model uses GPT to generate accurate text descriptions of the image’s concept from unlabeled images for training. While it seems similar to our proposed method in its use of inversion and Knowledge Distillation (KD) techniques, the SEARLE model employs inversion to obtain concept embeddings of images and simplifies this process using KD.

- The SEARLE model shows lower performance on average compared to other models. This is likely because the SEARLE model is designed to maintain image-level concepts and transform contextual information. The concept at this time is mostly the main instance present in the image. Our proposed model, which maintains the contextual information of the image and performs CIR based on human-defined image-level concepts, differs from the SEARLE model that transforms contextual information.

This explanation is noted in the Related Work section Page 6, Lines 137-141, and Comparison with CoIR models section Page 17, Lines 423-429.

C7: In the Methodology section, the use of knowledge distillation requires further discussion and presentation of more detailed specifics.

R7: Thank you for advising us to provide a detailed description of the knowledge distillation method used in the paper. We have thoroughly revised the knowledge distillation section in the Methodology to include a more comprehensive explanation. In addition to a general explanation of knowledge distillation, we have detailed the implementation and training of knowledge distillation in this study.

C8: Details regarding the training of the model have not been mentioned. 

R8: Thank you for your advice on describing the training process of the proposed model in the paper. We have added detailed explanations of the training methods for the Overall architecture, backward search, and the knowledge distillation model. For the Overall architecture, we reorganized the order of the paragraphs to make the training process of the proposed model easier to understand. The backward search has been explained more clearly by adding pseudocode to illustrate the embedding learning process as shown in Fig 3. For the knowledge distillation model, we have provided a more detailed description of the training process in the knowledge distillation section.

---

## [Decision Letter · Decision Letter 1]

10 Jul 2024

PONE-D-24-16658R1Backward Inductive Deep Neural Image SearchPLOS ONE

Dear Dr. Kim,

Thank you for submitting your manuscript to PLOS ONE. After careful consideration, we feel that it has merit but does not fully meet PLOS ONE’s publication criteria as it currently stands. Therefore, we invite you to submit a revised version of the manuscript that addresses the points raised during the review process.

We look forward to receiving your revised manuscript.

Kind regards,

Dr. S. M. Anas, Ph.D.(Structural Engg.), M.Tech(Earthquake Engg.)

Academic Editor

PLOS ONE

Journal Requirements:

Additional Editor Comments:

Dear Authors,

Thank you for your revised submission of the manuscript entitled "Backward Inductive Deep Neural Image Search" [PONE-D-24-16658R1]. The manuscript has been sent to peer reviewers for reevaluation.

One reviewer is satisfied with your responses, while the other has suggested some minor revisions. Based on the reviewers' comments and my preliminary assessment, I have decided to take a Minor Revision decision, subject to the approval of the editorial board.

Please submit a revised manuscript along with a detailed point-by-point response to the reviewers' comments.

We look forward to receiving your revised manuscript soon.

Best regards,

Dr. S. M. Anas

Academic Editor

PLOS ONE

Reviewers' comments:

Reviewer's Responses to Questions

**Comments to the Author**

1. If the authors have adequately addressed your comments raised in a previous round of review and you feel that this manuscript is now acceptable for publication, you may indicate that here to bypass the “Comments to the Author” section, enter your conflict of interest statement in the “Confidential to Editor” section, and submit your "Accept" recommendation.

Reviewer #1: All comments have been addressed

Reviewer #2: All comments have been addressed

2. Is the manuscript technically sound, and do the data support the conclusions?

Reviewer #1: Yes

Reviewer #2: Yes

3. Has the statistical analysis been performed appropriately and rigorously? 

Reviewer #1: No

Reviewer #2: Yes

4. Have the authors made all data underlying the findings in their manuscript fully available?

Reviewer #1: Yes

Reviewer #2: Yes

5. Is the manuscript presented in an intelligible fashion and written in standard English?

Reviewer #1: Yes

Reviewer #2: Yes

6. Review Comments to the Author

Reviewer #1: The article is in good shape now.

The following point need to be addressed as a minor revision

1. I would suggest the title “Backward Induction-based Deep Image Search”. It is not mandatory to follow this suggestion. I leave it to the worthy Editor to check and accept the article.

Reviewer #2: (No Response)

7. PLOS authors have the option to publish the peer review history of their article (what does this mean?). If published, this will include your full peer review and any attached files.

Reviewer #1: **Yes: **Prof Dr Shahzad Ahmad Qureshi

Reviewer #2: No

---

## [Author Response · Author response to Decision Letter 1]

2 Aug 2024

To Academic Editor 

C1: Please review your reference list to ensure that it is complete and correct. If you have cited papers that have been retracted, please include the rationale for doing so in the manuscript text, or remove these references and replace them with relevant current references. Any changes to the reference list should be mentioned in the rebuttal letter that accompanies your revised manuscript. If you need to cite a retracted article, indicate the article’s retracted status in the References list and also include a citation and full reference for the retraction notice. 

R1: 

Thank you for your advice on the completeness of the reference list. We reviewed and revised the reference information. The main changes we made are as follows: (We apologize for the untracked changes in the reference section, as we used Endnote for reference management.)

- Revised the journal/conference names to their official titles. (e.g. Association for the Advancement of Artificial Intelligence revised to AAAI Conference on Artificial Intelligence)

- Some papers have been updated on arXiv, resulting in multiple citation versions exists. We updated the citations to reflect the final versions of the papers.

- We searched and added the page numbers where the respective references are in the journal or proceedings.

We also carefully searched whether any of the references have been retracted. To verify the retraction status, we checked the respective journal websites and cross-checked with http://retractiondatabase.org/, www.webofscience.com, and www.semanticscholar.org. As a result, we confirmed that there are no retracted papers in the reference list. For papers published on arXiv, we confirmed that they are still publicly available. Please let us know if there are any issues with the references that we have not noticed, and we will make the necessary corrections.

To Reviewer 1

C1: I would suggest the title “Backward Induction-based Deep Image Search”. It is not mandatory to follow this suggestion. I leave it to the worthy Editor to check and accept the article.

R1: Thank you for your advice on revising the title of our paper. As you suggested, we have changed it to "Backward Induction-based Deep Image Search" to better emphasize our proposed method.

---

## [Decision Letter · Decision Letter 2]

26 Aug 2024

Backward Induction-based Deep Image Search

PONE-D-24-16658R2

Dear Dr. Kim,

We’re pleased to inform you that your manuscript has been judged scientifically suitable for publication and will be formally accepted for publication once it meets all outstanding technical requirements.

Kind regards,

Jin Liu

Academic Editor

PLOS ONE

Additional Editor Comments (optional):

This manuscript can be accepted now

Reviewers' comments:

Reviewer's Responses to Questions

**Comments to the Author**

1. If the authors have adequately addressed your comments raised in a previous round of review and you feel that this manuscript is now acceptable for publication, you may indicate that here to bypass the “Comments to the Author” section, enter your conflict of interest statement in the “Confidential to Editor” section, and submit your "Accept" recommendation.

Reviewer #1: All comments have been addressed

Reviewer #2: All comments have been addressed

2. Is the manuscript technically sound, and do the data support the conclusions?

Reviewer #1: Yes

Reviewer #2: Yes

3. Has the statistical analysis been performed appropriately and rigorously? 

Reviewer #1: (No Response)

Reviewer #2: Yes

4. Have the authors made all data underlying the findings in their manuscript fully available?

Reviewer #1: Yes

Reviewer #2: Yes

5. Is the manuscript presented in an intelligible fashion and written in standard English?

Reviewer #1: Yes

Reviewer #2: Yes

6. Review Comments to the Author

Reviewer #1: All my suggestions have been thoroughly addressed. I am satisfied with this article without any further modifications.

The article is in good shape, and it may be accepted.

Reviewer #2: (No Response)

7. PLOS authors have the option to publish the peer review history of their article (what does this mean?). If published, this will include your full peer review and any attached files.

Reviewer #1: No

Reviewer #2: No

---

## [Editor Report · Acceptance letter]

29 Aug 2024

PONE-D-24-16658R2 

PLOS ONE

Dear Dr. Kim, 

I'm pleased to inform you that your manuscript has been deemed suitable for publication in PLOS ONE. Congratulations! Your manuscript is now being handed over to our production team.

Kind regards, 

on behalf of

Professor Jin Liu 

Academic Editor

PLOS ONE